# Assessing the Immunomodulatory Effect of Size on the Uptake and Immunogenicity of Influenza- and Hepatitis B Subunit Vaccines In Vitro

**DOI:** 10.3390/ph15070887

**Published:** 2022-07-18

**Authors:** Rick Heida, Philip A. Born, Gabriela Tapia-Calle, Henderik W. Frijlink, Anna Salvati, Anke L. W. Huckriede, Wouter L. J. Hinrichs

**Affiliations:** 1Department of Pharmaceutical Technology and Biopharmacy, University of Groningen, 9713 AV Groningen, The Netherlands; r.heida@rug.nl (R.H.); philipborn@gmail.com (P.A.B.); h.w.frijlink@rug.nl (H.W.F.); 2Department of Medical Microbiology and Infection Prevention, University of Groningen, University Medical Center Groningen, 9713 AV Groningen, The Netherlands; gab.tapiac@gmail.com (G.T.-C.); a.l.w.huckriede@umcg.nl (A.L.W.H.); 3Department of Nanomedicine and Drug Targeting, University of Groningen, 9713 AV Groningen, The Netherlands; a.salvati@rug.nl

**Keywords:** dendritic cells, drug delivery systems, flow cytometry, hepatitis B virus, human peripheral blood mononuclear cells, influenza virus, in vitro, microparticles, nanoparticles, subunit vaccines

## Abstract

Viral subunit vaccines are a safer and more tolerable alternative to whole inactivated virus vaccines. However, they often come with limited efficacy, necessitating the use of adjuvants. Using free and particle-bound viral antigens, we assessed whether size affects the uptake of those antigens by human monocyte-derived dendritic cells (Mo-DCs) and whether differences in uptake affect their capacity to stimulate cytokine production by T cells. To this end, influenza antigens and hepatitis B surface antigen (HBsAg) were covalently conjugated to polystyrene particles of 500 nm and 3 μm. Cellular uptake of the antigens, either unconjugated or conjugated, and their capacity to stimulate T cells within a population of human peripheral blood mononuclear cells (PBMCs) were measured by flow cytometry. Conjugation of both antigens to particles significantly increased their uptake by Mo-DCs. Moreover, both the 500 nm and 3 μm influenza conjugates induced significantly higher numbers of cytokine-producing CD4^+^ T cells and induced increased production of the pro-inflammatory cytokines IFNγ and TNFα. In contrast, conjugation of HBsAg to particles did not notably affect the T cell response. In conclusion, conjugation of antigen to 500 nm and 3 μm particles leads to increased antigen uptake by human Mo-DCs, although the capacity of such conjugates to induce T cell stimulation likely depends on the immunological status of the PBMC donor.

## 1. Introduction

As opposed to highly immunogenic vaccine formulations like live attenuated and whole inactivated pathogens, subunit vaccines only consist of one or more antigens against which an immune response must be elicited. Although these protein-based vaccines generally have a better safety profile, their immunogenicity is often limited due to the lack of pathogen-associated molecular patterns and the induction of immunological tolerance [1]. Therefore, an important challenge in vaccine development remains to create vaccines that are safe but at the same time sufficiently immunogenic to generate a strong and long-lasting immune response [2]. In this respect, a vaccine’s effectiveness is partially dependent on its ability to be internalized by antigen-presenting cells (APCs). APCs are essential for the endo- or phago-lysosomal processing and initial presentation of antigens to naïve CD4^+^ T cells via MHC-II complexes [3]. Particulate delivery systems may improve the uptake of subunit-based vaccines. They can act as immune-potentiating adjuvants by mimicking the physical characteristics of the pathogen, i.e., its size, surface charge, shape, and rigidity [2,4,5]. By this means, particulate delivery systems, using particles to deliver antigens that are either encapsulated in or conjugated to the particle, can favor antigen uptake by APCs. For this reason, they have gained significant attention over the last decades [6,7].

A key parameter that determines the efficacy of particulate delivery systems is their size. Previous studies, however, show conflicting results regarding the optimal size range for the induction of an efficient and long-lasting immune response, with claims favoring both smaller-sized particles over bigger ones and vice versa [8,9,10,11,12,13,14,15]. Also, the majority of these studies have been performed in animal models which have shown not always to be reflective of the human situation [16,17,18]. In this respect, a more representative model could be found in human-derived dendritic cells. These cells, having a highly phagocytic nature, continuously scan their environment for foreign antigens and are therefore considered the gatekeepers of the human immune system [19]. By bridging the innate and adaptive immune system, they are essential to vaccine immunogenicity. While a limited number of studies have used human dendritic cells to assess the effect of particle size on uptake [20,21,22], these studies used particles encapsulating antigens to improve their delivery, while particles with antigens conjugated on their outer surfaces, to the best of our knowledge, have not been tested as of yet on human dendritic cells. Similarly, the effect of particle size on the vaccine-induced immune response by human T cells has not yet been elucidated.

Therefore, in the present study, we made use of influenza subunits (hereafter referred to as influenza antigens) and hepatitis B surface antigen (HBsAg) and conjugated them covalently to either polystyrene nanoparticles of 500 nm or microparticles of 3 µm in order to increase their size. By this means, we first aimed to determine the effect of the conjugation on antigen uptake by human monocyte-derived dendritic cells (Mo-DCs) compared to free (unconjugated) antigens, using flow cytometry. Secondly, we assessed whether modulating the particle size affected downstream T cell responses in unfractionated cultures of human peripheral blood mononuclear cells (PBMCs) by measuring intracellular cytokine expression.

## 2. Results

### 2.1. Conjugates and Conjugation Efficiencies

Three types of conjugates were used in this study (depicted in Figure 1). Firstly, unlabeled influenza antigens and HBsAg were conjugated to fluorescently labeled carboxyl-functionalized particles (Figure 1a). These conjugates were used to determine uptake into human Mo-DCs by fluorescence microscopy. Secondly, fluorescently labeled antigens conjugated to unlabeled amino-functionalized particles were used for flow-cytometric analysis to quantify the uptake by human Mo-DCs (Figure 1b). Lastly, unlabeled antigens conjugated to unlabeled amino-functionalized particles (Figure 1c) were used to assess T cell cytokine expression from within human PBMCs. The degree of conjugation was determined by protein assays. By subtracting the amount found in the supernatant after the conjugation procedure from the amount that was initially added, we could indirectly determine the amount of antigen that was successfully conjugated to the particles (Table 1). The results showed that in all cases, conjugation was achieved; however, the conjugation efficiency was very different for the two particle sizes and different antigens. Using fluorescence microscopy, we could visually confirm successful conjugation (Figure 2).

### 2.2. Conjugate Particle Size and Zeta Potential

The particle size before and after conjugation was determined by dynamic light scattering (DLS) and laser diffraction analysis (Figure 3). DLS showed that the hydrodynamic size of the non-conjugated 500 nm amino-functionalized particles was indeed approximately 500 nm in diameter (502 ± 5 nm). After conjugation with influenza antigens, the average hydrodynamic diameter of the particles increased (642 ± 10 nm). A similar effect was observed upon conjugation with HBsAg (596 ± 9 nm). Similarly, laser diffraction analysis showed that the size of the non-conjugated 3 µm particles was indeed approximately 3 µm in diameter (3020 ± 0 nm). After conjugation with influenza antigens, the average geometrical diameter of the conjugated particles increased slightly (3100 ± 0 nm). A similar effect was observed with HBsAg (3160 ± 0 nm) (Figure 3a).

Regarding their zeta potential, the non-conjugated 500 nm amino-functionalized polystyrene particles showed a negative value when dispersed in PBS (−49 ± 1 mV). This suggests that, under these conditions, the amino groups were not protonated. After conjugation with influenza antigens or HBsAg, the zeta potential became less negatively charged (−37 ± 1 mV and −36 ± 0 mV, respectively) (Figure 3b). As a rule of thumb, colloidal formulations are considered stable when the zeta potential is lower than −30 mV or higher than +30 mV [23]. Therefore, these results suggest that at physiological conditions (PBS), colloidally stable conjugates were formed. Indeed, no signs of aggregation were observed during storage. In addition, the polydispersity index of each sample determined by DLS was 0.0, which indicates that the particle size distribution was narrow.

### 2.3. Visualization of Antigen Uptake

In order to confirm uptake by the Mo-DCs, cells that had been exposed to the different formulations (either fluorescently-labeled particles with or fluorescently-labeled particles without unlabeled antigen) were analyzed by wide-field fluorescence microscopy. Figure 4 shows that both the unconjugated fluorescent particles (Figure 4a,c; without antigen) and the fluorescent particles (Figure 4b,d) with conjugated influenza antigens were internalized by Mo-DCs.

### 2.4. Quantification of Antigen Uptake in Mo-DCs

In order to determine how conjugation of antigens to particles affected their uptake by Mo-DCs, we used FITC-labeled antigens, either conjugated to particles or unconjugated (free), and compared their uptake over time by flow cytometry. Because of the very different conjugation efficiencies (Table 1), it was not possible to expose cells of different groups to the same amounts of antigen (either free or conjugated to the two types of particles) without exposing cells to very high doses of particles. Therefore, these experiments were performed with an equal number of particles as determined in a pilot experiment using THP-1 cells (Appendix A). Here, the fluorescence was compared for three different amounts of particles (20, 50, and 100 µg). Based on this, we determined for both particle sizes that exposure to 20 µg of particles gave enough signal to easily quantify cell uptake kinetics over time without overflowing the cells with particles. In order to enable comparison within each group between the uptake of free antigen and particle-conjugated antigens within each group, the amount of free antigen corresponded to the amount that was bound to the particles.

Figure 5 shows the average fluorescence of the Mo-DCs from four donors over time after exposure to either the unconjugated or the conjugated antigens (gating strategy and scatter plots for each individual donor can be found in Appendix A). In Appendix A, the overlays of some of the corresponding cell fluorescence distributions are shown for one representative donor. Interestingly, in the cells exposed to the 3 µm particles, because of their larger size (thus high fluorescence per particle), separate peaks can be distinguished, corresponding to different amounts of particles being taken up (Appendix Aa,e). Overall, the results show that uptake increased over time. In all cases, conjugation of the antigens to particles resulted in higher antigen amounts inside cells as compared to stimulation with free antigen. Interestingly, for the cells exposed to HBsAg-conjugated particles (5b and 5d), that contained more antigens per surface area, the differences in uptake between particle-conjugated and free antigen were even more profound.

Due to the use of different antigen amounts for the uptake experiments with 500 nm and 3 μm particles as outlined above, it was not possible to directly compare the cell fluorescence intensity values between sizes. However, the slopes extracted from a linear fit of the uptake kinetics can be used as an estimation of the uptake rate (Table 2), which can then be used to compare the uptake rates of free and particle-conjugated antigens. In this way, we found that the uptake rate was approximately 2.6 times and 4.2 times higher for the 3 µm and 500 nm HBsAg conjugates, respectively, than the uptake rate of the free antigen. This confirmed once more that conjugation of antigens to particles increases their uptake by Mo-DCs. The stronger effect on uptake rate observed for the 500 nm HBsAg conjugates is probably connected to the higher uptake efficiency of smaller nanoparticles in comparison to the larger microparticles. However, this was not the case for the influenza conjugates where, despite the much lower amount of antigen per particle and the larger size (usually associated to lower uptake efficiency), the increase in uptake rate was more pronounced for the larger 3 µm influenza antigen conjugates than for the smaller conjugates (up to 18× uptake rate increase for the 3 µm with respect to the uptake rate of the free antigen, while only a 1.2× uptake rate increase for the 500 nm conjugates was observed). This suggests that in the case of influenza antigens conjugated to microparticles, even low levels of bound antigen already increase their uptake. In conclusion, the results clearly show that conjugation of antigens to particles of different sizes increases antigen uptake.

### 2.5. Assessing T Cell Stimulation in a Population of PBMCs

To elucidate whether stimulation of uptake also translated into a higher degree of T cell activation and cytokine expression, unfractionated PBMCs were stimulated for 10 days with plain antigens, particle-conjugated antigens, or with unconjugated particles to allow antigen-specific T cells to proliferate and differentiate into cytokine-producing cells. Thereafter, the cells were harvested for intra-cellular cytokine staining. In this way, we observed that free influenza antigens, although not significant, elicited the activation of antigen-specific T cells, reflected by higher frequencies of CD4^+^ and CD8^+^ T cells producing IFNγ (Figure 6a,g), TNFα (Figure 6b,h), and IL-10 (Figure 6c,i), compared to cell cultures stimulated with unconjugated particles. Also, the integrated median fluorescence intensity (iMFI), which represents the total amount of cytokine being produced, was higher for CD4^+^ and CD8^+^ T cells producing IFNγ (Figure 6d,j) and TNFα (Figure 6e,k), and for CD8^+^ T cells producing IL-10 (Figure 6l), in cultures exposed to the free antigen than in those stimulated with unconjugated particles.

Interestingly, both the exposure of PBMCs to 500 nm and 3 µm influenza conjugates resulted in a significant increase in the frequency of CD4^+^ T cells producing IFNγ and TNFα (Figure 6a,b) and in total amounts of these cytokines (Figure 6d,e) compared to exposure of PBMCs to unconjugated particles. On the contrary, this was not the case for IL-10 (Figure 6c,f), which may imply down-regulation of this anti-inflammatory cytokine. In the CD8^+^ T cell subset, though, only the frequency of cells expressing IL-10 was significantly increased for both conjugate sizes (Figure 6i), while TNFα-expressing cells were only significantly increased upon stimulation with the 3 µm conjugates.

The T cell response to the HBsAg, either plain or conjugated to 500 nm or 3 µm particles, was generally very poor compared to the response to the influenza formulations, as reflected in both lower frequencies of cytokine producing cells and lower total cytokine amounts (Figure 7).

## 3. Discussion

In this study, we assessed whether modulating the size of viral antigens, by conjugation to nano- and microparticles, would alter their uptake into human-derived Mo-DCs and whether this would subsequently influence cytokine expression in T cells within a population of human PBMCs. We observed that conjugated antigens were taken up with greater efficiency than their unconjugated counterparts. Interestingly though, more efficient uptake only translated to significantly enhanced T cell activation upon stimulation with the influenza, and not the HBsAg conjugates. Specifically, influenza antigens conjugated to both 3 µm and to 500 nm particles induced higher levels of the pro-inflammatory cytokines IFNγ and TNFα in CD4^+^ T cells than the unconjugated particles. This indicates successful processing of the antigens by antigen-presenting cells and subsequent presentation to T cells present in the PBMC culture. Earlier research from our group demonstrated that the majority of T cells in PBMC cultures responding to influenza antigen carry the memory T cell marker CD45RA and are thus derived from earlier exposure to influenza virus [24]. Memory T cells have a relatively low threshold of T cell receptor activation and do not require activated antigen-presenting cells providing co-stimulation [25]. Conjugation of HBsAg to particles did not result in more efficient T cell activation than unconjugated HBsAg, as CD4 and particularly CD8 responses to HBsAg were generally low. This probably reflects the fact that our donors were most likely naïve to hepatitis B virus and thus did not have memory responses to HBsAg. Naïve T cells have a higher threshold for T cell receptor activation than memory T cells and are more dependent on co-stimulation [25]. Thus, while coupling of influenza antigens and HBsAg to particles increased the uptake of both antigens, we believe that this only improved stimulation of influenza-specific memory T cells but not naïve HBsAg-specific T cells, as proper co-stimulatory signals required for activation of naïve T cells were lacking. Future studies should check on the production of these co-stimulatory signals to verify this hypothesis.

When looking at particle size, it is relevant to note that dendritic cells preferentially process antigen-functionalized particulates at a size range up to 200 nm for the induction of a CD8^+^ T-cell-mediated immune response, which is important for elimination of virus-infected cells [26]. Thus, especially in the context of a multi-dimensional PBMC culture, it is likely that a part of the larger conjugates used in this study has been taken up by other APCs than dendritic cells, such as macrophages, in line with previous work by Champion et al. [27]. This may also partly explain the lacking response upon stimulation with HBsAg conjugates, as uptake by dendritic cells is essential for the initial priming of CD4^+^ T helper cells that are naïve for hepatitis B virus. Therefore, the T cells likely have not been activated to produce cytokines. Another reason for the low cytokine production may lay with the mechanisms by which particles of different sizes are internalized, both with respect to the uptake of free antigens as well as to intracellular antigen trafficking and distribution. Specifically, different routes of uptake, i.e., phagocytosis versus receptor-mediated endocytosis, could have influenced the processing of antigens and subsequent presentation on MHC complexes [26]. A thorough characterization of the molecular details of these processes would be required in order to determine the optimal particle properties for achieving a stronger immune response.

Altogether, the relation between size and the induced immune response may not be set in stone and is probably also dependent on several other factors, such as charge [20,22], surface modification [22,28], shape, and rigidity [29,30], as comprehensively reviewed by Slütter and coworkers [5,26]. The next steps in order to further understand the role of size in antigen uptake and processing would be to extend these studies to particles in the low- to medium- nanometer size range, i.e., up to 200 nm as those are more reflective of the general size range of viruses [26]. Additionally, it would be important to develop methods that would allow control of antigen conjugation efficacy. Preparing particles at increasing amounts of antigen per particle would allow disentanglement of effects due to particle size and effects caused by variations in antigen density. In addition, with regard to the covalent conjugation of antigens to particles and fluorescent labeling of the antigen, the extent to which these modifications themselves influence downstream processing of antigens in endo- or phago-lysosomal compartments should be assessed, as this likely affects their immunogenicity as well. Obviously, we do not envision clinical use of polystyrene particles, as they are nondegradable and may lead to toxic effects. Therefore, it would be important to use bio-compatible and -degradable particles, such as poly(lactic-co-glycolic acid)-based particles, for future studies, as opposed to the polystyrene particles used here [7].

In conclusion, defining the most relevant characteristics of subunit vaccine formulations for the effective stimulation of human dendritic cells is required to further increase our understanding of the potential and the optimal properties of vaccine delivery systems and their size. While the current study showed that linking antigens to particles of 500 nm and 3 µm can only partly influence the immune response based on cytokine expression in T cells, particle-conjugated antigens may still indirectly lead to a more profound response by increasing the rate of uptake and by increasing antigen visibility in vivo. Future studies should look into the possibility of translating the work described here using biodegradable polymers along a broader spectrum of size and see if these aid in uptake, preferably by several cell types. For in vivo studies, the route of administration should not be neglected as it is an important component of effective vaccine delivery.

## 4. Materials and Methods

### 4.1. Influenza Subunit Production

For the production of influenza subunits, 0.6 mg/mL of Tween 80 and 3.0 mg/mL of cetrimonium bromide were added to whole inactivated influenza virus with a total protein concentration of 0.8 mg/mL (X-31, a conventionally produced reassortant of A/Aichi/68 (H3N2) and A/PR/8/34, NIBSC, Potters Bar, UK). The suspension was then slowly rotated for 3 h at 4 °C. Subsequently, the mixture was centrifuged at 50,000 rpm (TLA100.3 rotor) for 30 min at 4 °C to remove the nucleocapsid. After transferring the subunit-containing supernatant to a new tube, detergents were removed by adding 634 mg/mL Amberlyte X AD-4 (Sigma-Aldrich, St. Louis, MO, USA) Biobeads to the supernatant, followed by overnight incubation under slow rotation at 4 °C. Protein content was determined using the Lowry assay [31]. In order to reach the desired concentration window needed for labeling the antigens, the subunit was subsequently concentrated using a Pierce^™^ Protein Concentrator PES (Thermo Fisher Scientific, Waltham, MA, USA) with a molecular weight cut-off of 100 kDa, as influenza hemagglutinin and neuraminidase antigens present fall above this range. The concentrated fraction was used for further experiments. The HBsAg (without adjuvants) was a generous gift from the Serum Institute of India (Pune, India).

### 4.2. Conjugation of Antigens to 3 µm and 500 nm Polystyrene Particles

#### 4.2.1. Fluorescent Particles for In Vitro DC Stimulation and Uptake Imaging

Influenza antigens were conjugated to fluorescent polystyrene particles in order to track uptake by Mo-DCs qualitatively, using fluorescence microscopy. Conjugation of the vaccine to fluorescent polystyrene particles was performed based on a slightly modified version of the manufacturer’s protocol. In short, 4.17 mg of Fluoresbrite Yellow-Green (YG) carboxyl-functionalized polystyrene particles of 3 µm and 500 nm (approximately 0.28 × 10^9^ and 60.7 × 10^9^ particles, respectively, Polysciences Europe GmbH, Hirschberg an der Bergstraße, Germany) were incubated for 15 min at room temperature in the presence of 6.67 µL (200 mg/mL) of freshly prepared carbodiimide in Polylink Coupling Buffer (50 mM MES, pH 5.2, 0.05% Proclin 300). Subsequently, 66.67 µg and 43.2 µg of influenza antigens were added to the 500 nm and 3 µm pre-activated particles, respectively, and then conjugated under slow overnight rotation at room temperature. After centrifugation at 15,000 rpm (max speed) for 6 min, the supernatants were collected for indirect determination of bound protein. The indirect measurement was chosen because direct measurements on protein–particle conjugates were described to lead to underestimations [32]. The conjugated fluorescent particles were then washed two times and resuspended in PBS 1x (pH 7.4) up to a concentration of 4 mg/mL before storing at 4 °C. A modified Pierce™ Coomassie Plus (Thermo Fisher Scientific) Assay (Thermo Fisher Scientific) was used according to manufacturer’s protocol to determine the conjugation efficiency, as other protein assays (including Lowry) were found to interfere with the carbodiimide reagent [33,34]. Samples were run in triplicate.

#### 4.2.2. Non-Fluorescent Particles for In Vitro PBMC Stimulation

For the immune response studies, conjugation of non-fluorescent influenza antigens or HBsAg to polystyrene particles was carried out according to manufacturer’s instructions. Shortly, in duplicate, 12.5 mg of 3 µm or 500 nm amino-functionalized particles (approximately 0.84 × 10^9^ and 182 × 10^9^ particles, respectively, Polysciences Europe) were incubated in 0.5 mL aqueous 8% (*v*/*v*) glutaraldehyde for 4 h at room temperature. After centrifugation of the particles at 15,000 rpm for 6 min, the glutaraldehyde-containing supernatant was discarded and 200 µg of influenza antigens or HBsAg were added to the pre-activated particles in 1 mL PBS. The conjugates were formed by overnight incubation at room temperature under gentle rotation. After centrifugation of the conjugated particles, the supernatant was carefully collected to determine unbound protein content using Lowry assay [31]. To block unreacted sites, the conjugated particles were resuspended in 1 mL of 0.2 M ethanolamine and slowly rotated for 30 min at room temperature. After removal of the ethanolamine by centrifugation, the conjugated particles were resuspended in PBS, pH 7.4.

#### 4.2.3. Conjugation of FITC-Labeled Antigens to Non-Fluorescent Particles for In Vitro Uptake Kinetics in Monocyte-Derived Dendritic Cells

In order to study uptake kinetics, we made use of fluorescently labeled antigens conjugated to non-fluorescent particles. The antigens were fluorescently labeled using a FITC Conjugation Kit (Fast)—Lightning-Link^®^ (Abcam, Cambridge, UK). The assay was performed according to the manufacturer’s protocol to yield 1 mg of labeled antigens. Hereafter, the FITC-labeled antigens were linked to 12.5 mg of non-fluorescent 3 µm and 500 nm amino-functionalized polystyrene particles using the method as described above for the non-labeled antigens. To every condition, 400 µg of FITC-labeled antigens (either influenza antigens or HBsAg) was added. Hereafter, the conjugation efficiency was determined with a modified Pierce™ Coomassie Plus Assay (Thermo Fisher Scientific). Conjugation was visually confirmed by microscopy, using a Deltavision^™^ Elite high-resolution fluorescence microscope (GE Healthcare UK Ltd., Little Chalfont, UK) equipped with a 60× oil immersion objective. Data acquisition was performed using the system-integrated GFP/mCherry filter setting in combination with the Deltavision SoftWoRx^™^ 6 acquisition and deconvolution software (GE Healthcare). The data were subsequently analyzed and processed using FIJI imaging software [35].

### 4.3. Determination of Size and Zeta Potential of the Conjugates

The hydrodynamic diameter (d_h_), polydispersity percentage, and zeta potential of the 500 nm non-fluorescent conjugated particles before and after conjugation were determined by a Mobius zeta potential and dynamic light scattering (DLS) detector, connected to an Atlas cell pressurization system (Wyatt Technology, Santa Barbara, CA, USA). The zeta potential determines the magnitude of the electrostatic repulsion between particles and is known to be one of the most important factors that affects colloidal stability. In triplicate, approximately 200 µL of conjugated particles (50 times diluted in PBS 1x) was injected into the Mobius cell via the Atlas injection port. To minimize air bubbles, the sample-containing Mobius cell was pressurized with approximately 15 Bar by the Atlas system. The hydrodynamic diameter and electrophoretic mobility were determined simultaneously using a laser with a wavelength of 532 nm and a detector angle of 163.5°. At least five scans were performed with an acquisition time of five seconds. The zeta potential was derived from the electrophoretic mobility (Smoluchowski model) using the Dynamics software. The size and zeta potential of the 3 µm particles could not be determined using this method due to settling of these relatively big particles. Therefore, the geometrical particle size (d_g_) of the non-fluorescent 3 µm conjugates was determined instead by laser diffraction analysis. In brief, 20 µL of the 3 µm particle suspension (1.68 × 10^9^ particles/mL) was dispersed in triplicate, under stirring, in 45 mL of ultrapure water in a 50 mL quartz cuvette. A parallel beam laser diffraction set-up (Helos/BF, Sympatec GmbH, Clausthal-Zellerfeld, Germany) with a 100 mm lens (range: 0.5/0.9–175 μm) recorded three single 10 s measurements with a 50 s pause in between. The mean geometrical particle size was calculated according to the Fraunhofer diffraction theory using the manufacturer’s software.

### 4.4. Visualization of Antigen Uptake by Monocyte-Derived Dendritic Cells

The uptake of the conjugates in human monocyte-derived dendritic cells was qualitatively assessed by fluorescence microscopy. Hereto, cryo-preserved PBMCs of healthy donors that were isolated from buffy coats obtained from the Dutch blood bank (Sanquin, Amsterdam, The Netherlands) were thawed and brought to a density of 40–50 million cells per mL according to standardized in-house procedures, as described previously [36]. Hereafter, monocytes were isolated from PBMCs by adherence. For this, approximately 2 × 10^6^ cells resuspended in 1 mL RPMI-1640 medium (Gibco Life technologies Co., Carlsbad, CA, USA), containing L-glutamine and HEPES and supplemented with 10% FCS and 1% penicillin/streptomycin, were added to fetal calf serum (FCS)-treated wells (24 well-plate; Corning Inc., Corning, NY, USA) containing pre-rinsed and autoclaved Menzel^™^ coverslips with a diameter of 12 mm (Thermo Fisher Scientific). After 2 h, the wells were extensively washed with medium to remove non-adherent cells. As the monocyte fraction of PBMCs is estimated to range between 7–15% [37], approximately 140,000 monocytes per well were obtained after washing (assuming 7% monocytes which all adhered). The cells were then cultured at 37 °C, 5% CO_2_ in 500 µL medium. For differentiation into Mo-DCs, the medium was supplemented with 500 U/mL IL-4 and 450 U/mL GM-CSF (ProSpec-Tany TechnoGene Ltd., Ness-Ziona, Israel). Fresh cytokines were added every 2–3 days. On day 6 after seeding, fluorescently labeled antigen preparations were added to the cells as follows. After discarding 300 µL of medium from each well of the 24-well plate, 10 µL of the 500 nm and 3 µm influenza antigen-conjugated particle preparations were mixed into 300 µL of medium before adding the suspensions to the corresponding wells. After adding the particle combinations to the wells, all conditions were mixed carefully by pipetting. The plate was then incubated for 20–24 h at 37 °C with 5% CO_2_. In order to visualize antigen and particle uptake, the cells were fixed using 4% formaldehyde in PBS (Alfa Aesar, Haverhill, MA, USA) and stained with Phalloidin-iFluor 594 (Abcam) to visualize actin filaments followed by staining with Hoechst 33342, trihydrochloride, and trihydrate (Invitrogen, Waltham, MA, USA) to visualize the nuclei. All staining procedures were performed by following the manufacturer’s protocols. After staining, the coverslips were mounted face down onto glass slides (Waldemar Knittel Glasbearbeitungs GmbH, Braunschweig, Germany) using 2.5 µL of SlowFade^™^ Diamond Antifade mounting medium (Invitrogen). In order to visualize the uptake of the particles, wide-field fluorescence microscopy was applied using the Deltavision^™^ Elite microscope (GE Healthcare) as described above. The data were subsequently analyzed and processed using FIJI imaging software [35]. Brightness and contrast were adjusted only to minimize background noise.

### 4.5. Quantification of Antigen Uptake in Monocyte-Derived Dendritic Cells

In order to quantitatively assess the effect of particle conjugation on the uptake of antigens and to validate if the antigens themselves influenced uptake, a flow-cytometry-based method was used to quantify uptake on the basis of differences in fluorescence intensities, a method which had been established previously [38]. On the basis of a pilot experiment in a THP-1 cell line, we selected 20 µg as the number of particles that gave enough signal to be measured over time with flow cytometry. Hereafter, we aimed to determine differences in the kinetics of uptake in Mo-DCs. Mo-DCs were acquired from PBMCs of four different donors using the procedure described previously. Depending on the specific yield of PBMCs from a single donor, 140,000 to 225,000 monocytes per well were seeded in three 24-well plates (1 plate per time-point). In order to assess the effect of the particles on the uptake of antigens, we used fluorescently labeled antigens conjugated to non-fluorescent particles. Cells were stimulated in duplicate with conjugates or plain antigens for either 1 h, 3 h, or 6 h according to the scheme in Table 3. The amount of free antigen used in the control group matched the amount of antigen that was bound to 20 µg of particles. After stimulation, medium was discarded and cells were harvested directly by applying 1 mL of cold FACS buffer (PBS 1x −/− containing 2% FCS and 5 mM EDTA) for 10 min. No washing steps were included to prevent loosely adherent or freely floating dendritic cells to be washed away. Because of the expected low number of Mo-DCs, cells from the duplicate wells were pooled into a single tube. The cells were then pelleted at 600 g for 5 min and buffer was discarded by decanting. Cells were then washed once with 2 mL PBS, centrifuged, and stained subsequently for 10 min at 4 °C with the following markers: anti-human-CD14-VioBlue^®^ (Miltenyi Biotec, Bergisch Gladbach, Germany) and Zombie NIR^™^ fixable viability dye (Biolegend Inc., San Diego, CA, USA). Dyes were diluted in 50 µL PBS at a respective dilution of 1:50 and 1:100 in PBS. Hereafter, cells were washed with 1 mL of FACS buffer and after centrifugation, cells were fixed for 10 min using 100 µL of IC fixation buffer (Invitrogen). After fixation, 1 mL of FACS buffer was added to stop the fixation process. Samples were stored overnight to proceed with staining the next day. Subsequently, the cells were centrifuged, buffer was discarded, and cells were stained for 10 min at 4 °C with additional surface marker antibodies anti-HLA-DR-PE-Vio^®^770 and anti-CD209 (DC-SIGN)-APC (both from Miltenyi Biotec), each at a dilution of 1:50 in PBS. Single-stain compensation controls were taken along for each dye during the first experiment. Unstained controls (not stimulated with any formulation) and stained controls were also included. Since the cytometer settings were kept constant for each experiment, compensations were set up only once and subsequently applied to all experiments. Samples were measured with a Novocyte Quanteon^™^ Flow Cytometer (Agilent Technologies, Santa Clara, CA, USA). For analysis, we made use of the NovoExpress^®^ software (Agilent Technologies).

### 4.6. Evaluation of the Antigen-Induced T Cell Response

Freshly thawed PBMCs were seeded at a density of 1 × 10^6^ in 1 mL of RPMI-1640 medium (L-glutamine, HEPES) supplemented with 10% FCS and 1% penicillin/streptomycin and rested overnight. Cultures were maintained at 37 °C, with 5% CO_2_. On day 1, cells were stimulated with different antigen formulations. For plain influenza antigens and HBsAg, 10 μg of protein in a total volume of 10 μL was added to the seeded cells. Conjugates were added to an amount that resembled 10 μg of bound antigen. On day 5, 50% of the medium was refreshed and on day 10, cells were harvested to assess T cell responses by flow cytometry. At 12 h before harvesting, 10 μg/mL of Brefeldin A (eBioscience™, Thermo Fisher Scientific) was added as a protein transport inhibitor. Cells were harvested with FACS buffer (1× PBS supplemented with 2% FCS and 1 mM EDTA) and then stained with a viability marker (Viobility 405/450, Miltenyi Biotec) for 15 min at room temperature. Washed cells were then fixed and permeabilized with BD Cytofix/Cytoperm Kit (BD Biosciences, Franklin Lakes, NJ, USA) according to the manufacturer’s instructions. Next, intracellular staining was performed using the following fluorescently labeled antibodies: anti-IFNγ-FITC, anti-TNFα-PE, and anti-IL10-APC (all from Miltenyi Biotec). Cells were then washed and stained for surface markers with the following fluorescently labeled antibodies: anti-CD3-Pacific Blue, anti-CD4-APCCy7, and anti-CD8-PerCPCy5 (all from Miltenyi Biotec). Cells were acquired with a FlowLogic (Miltenyi Biotec).

## 5. Conclusions

This study shows that conjugation of influenza- and hepatitis B subunit vaccines to polystyrene particles leads to higher uptake rates in Mo-DCs although only uptake of influenza conjugates stimulates cytokine production by T cells.

## Figures and Tables

**Figure 1 pharmaceuticals-15-00887-f001:**
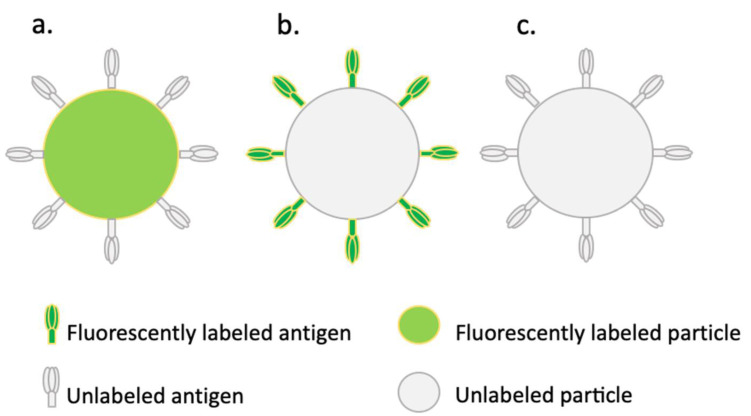
Schematic overview of the different conjugate types used in this study. (**a**) Unlabeled antigen conjugated to fluorescently labeled carboxyl-functionalized polystyrene particles used for fluorescence microscopy. (**b**) Fluorescently labeled antigen conjugated to unlabeled amino-functionalized polystyrene particles used for uptake quantification with flow cytometry. (**c**) Unlabeled antigen conjugated to unlabeled amino-functionalized polystyrene particles for assessing the immune response in a population of human PBMCs. All conjugate types were made with both 500 nm and 3 µm polystyrene particles.

**Figure 2 pharmaceuticals-15-00887-f002:**
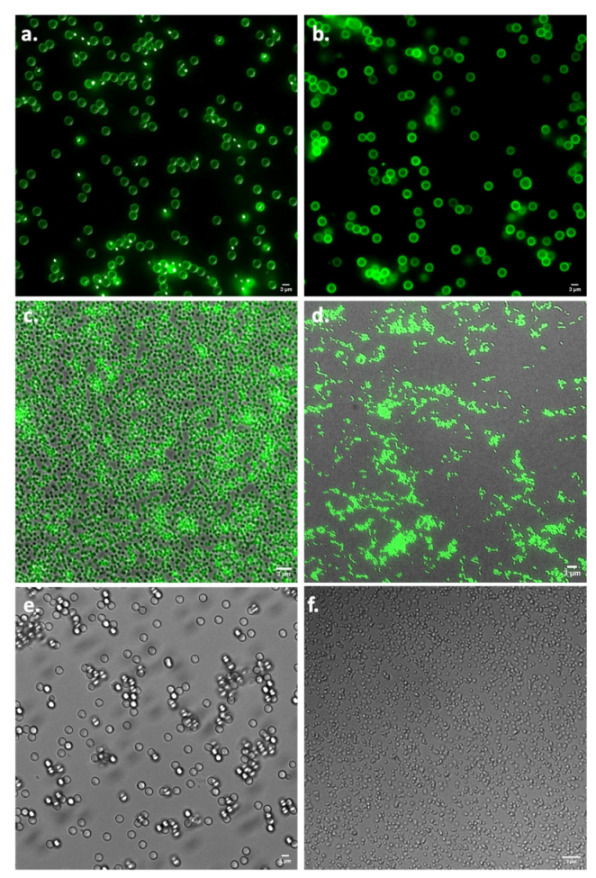
Visual conformation of successful conjugation of antigens to particles. (**a**) Conjugation of FITC-labeled influenza antigens to 3 µm particles. (**b**) Conjugation of FITC-labeled HBsAg to 3 µm particles. (**c**) Conjugation of FITC-labeled influenza antigens to 500 nm particles. (**d**) Conjugation of FITC-labeled HBsAg to 500 nm particles. (**e**) Unconjugated 3 µm particles. (**f**) Unconjugated 500 nm particles. Bar is 3 µm.

**Figure 3 pharmaceuticals-15-00887-f003:**
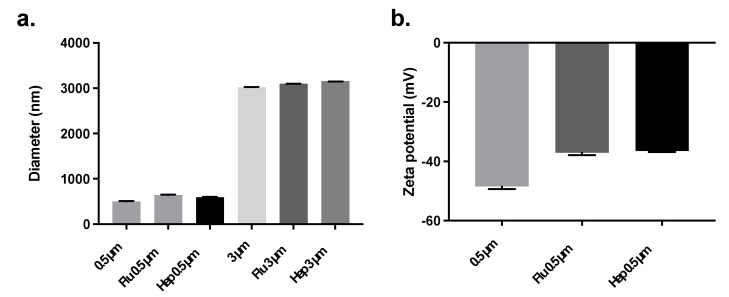
Diameter (**a**) and zeta potential (**b**) of polystyrene particles without and with conjugated influenza antigens (Flu) or hepatitis B surface antigen (Hep). The average hydrodynamic diameter and zeta potential of the 500 nm-conjugated particles were determined by a Mobius zeta potential and DLS detector. The average geometrical diameter of the 3 µm particles was determined by a Helos/BF parallel beam laser diffraction set-up (*n* = 3, mean ± SD). All particles were suspended in PBS, pH 7.4.

**Figure 4 pharmaceuticals-15-00887-f004:**
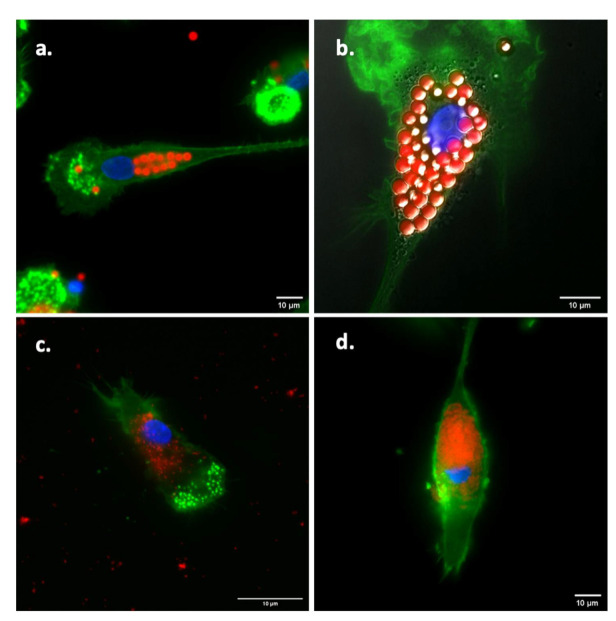
Uptake of nano- and microparticles by monocyte-derived dendritic cells (Mo-DCs). Mo-DCs were incubated for 20–24 h at 37 °C (5% CO_2_) with unconjugated fluorescently labeled particles (**a**,**c**) or influenza antigen-conjugated (**b**,**d**) particles with a diameter of 3 µm (**a**,**b**) and 500 nm (**c**,**d**). Cells were then fixed and stained, for filamentous actin using Phalloidin-iFluor 594 (Abcam, Cambridge, UK), and for nuclei using Hoechst 33342, trihydrochloride, and trihydrate (Invitrogen, Waltham, MA, USA), and subsequently examined using a Deltavision™ Elite high-resolution fluorescence microscope (GE Healthcare, Chicago, IL, USA). Actin filaments are shown in green, the nucleus in blue, and the particles, either conjugated or conjugated, in red. Bar is 10 µm.

**Figure 5 pharmaceuticals-15-00887-f005:**
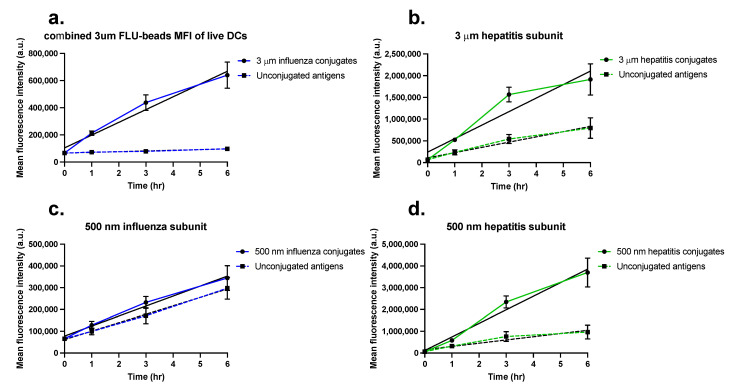
Mean fluorescence intensity (MFI) of dendritic cells over time upon stimulation with FITC-labeled antigen conjugates versus unconjugated antigens. (**a**) 3 μm FITC-labeled influenza antigen conjugates versus unconjugated FITC-labeled antigens. (**b**) 3 μm FITC-labeled HBsAg conjugates versus unconjugated FITC-labeled HBsAg. (**c**) 500 nm influenza antigen conjugates versus unconjugated FITC-labeled antigens. (**d**) 500 nm HBsAg conjugates versus unconjugated FITC-labeled HBsAg. Filled lines represent the MFI over time of cells stimulated with the FITC-labeled conjugates; dashed lines represent the MFI over time of cells stimulated with unconjugated FITC-labeled antigens. The black lines were used to calculate the slope of MFI over time using linear regression analysis. Data points represent the average MFI of four donors ± SEM.

**Figure 6 pharmaceuticals-15-00887-f006:**
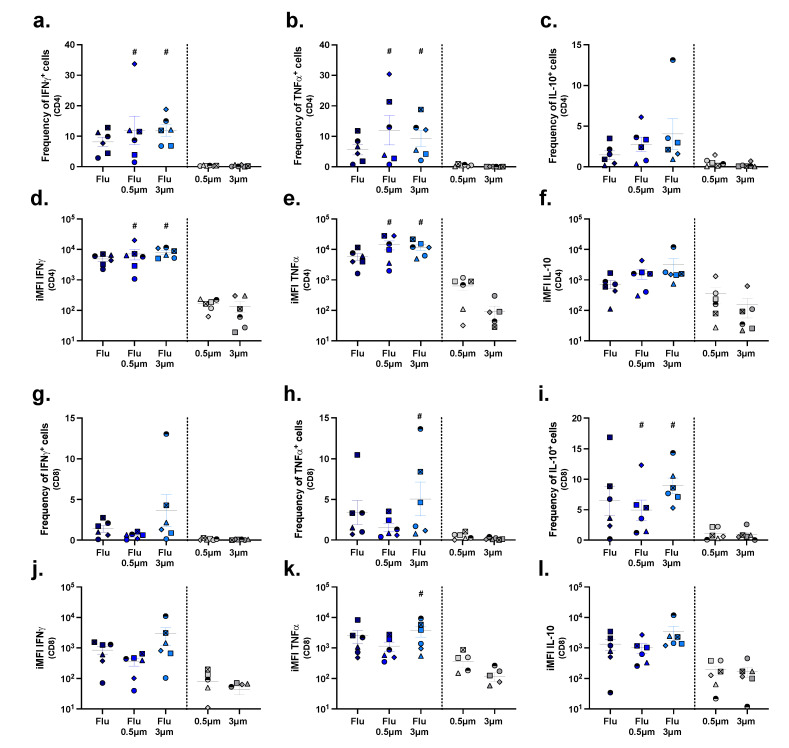
T cell responses upon stimulation with influenza antigen conjugates versus unconjugated antigens. Human PBMCs were stimulated with either unconjugated influenza antigens (Flu), influenza antigens conjugated to 500 nm particles (Flu-0.5 μm), or influenza antigens conjugated to 3 µm particles (Flu-3 μm). Cells from the control condition were stimulated with either unconjugated 500 nm or 3 µm particles. After 10 days, cells were harvested and evaluated by multicolor flow cytometry. Depicted are the frequencies of IFNγ-, TNFα-, and IL-10-producing CD4^+^ (**a**–**c**) and CD8^+^ (**g**–**i**) T cells and the respective integrated mean fluorescence intensities (iMFIs) ((**d**–**f**,**j**–**l**), respectively). Each symbol represents one donor (*n* = 6). Colors correspond to the different treatments as shown on the x-axis. Statistical analysis was performed using a one-way ANOVA, followed by a Tukey test. Significant differences (*p* < 0.05) between particle-conjugated antigens and unconjugated particles are represented with #.

**Figure 7 pharmaceuticals-15-00887-f007:**
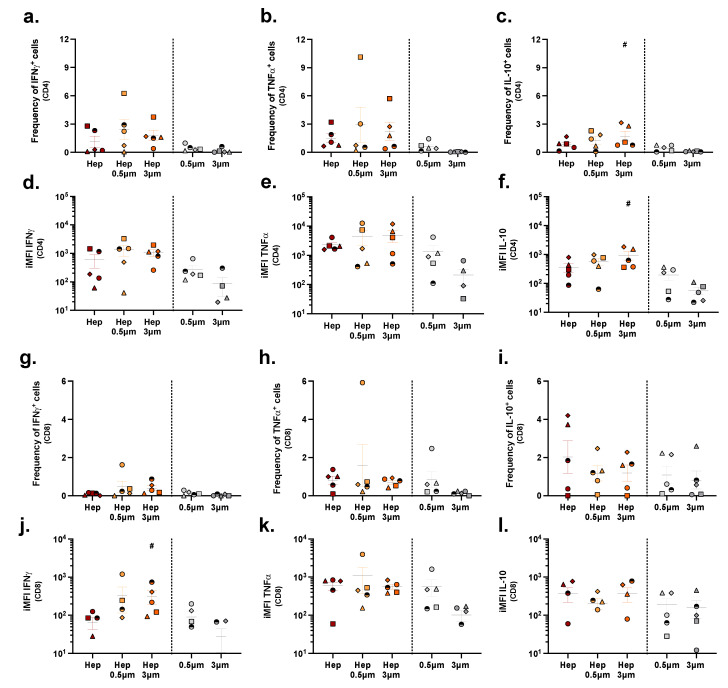
T cell responses upon stimulation with HBsAg conjugates versus unconjugated antigens. Human PBMCs were stimulated with either unconjugated HBsAg (Hep), HBsAg conjugated to 500 nm particles (Hep-0.5 μm), or HBsAg conjugated to 3 µm particles (Hep-3 μm). Cells from the control condition were stimulated with either unconjugated 500 nm or 3 µm particles. After 10 days, cells were harvested and evaluated by multicolor flow cytometry. Depicted are the frequencies of IFNγ-, TNFα-, and IL-10-producing CD4^+^ (**a**–**c**) and CD8^+^ (**g**–**i**) T cells and the respective iMFIs ((**d**–**f**,**j**–**l**), respectively). Each symbol represents one donor (*n* = 6). Colors correspond to the different treatments as shown on the x-axis. Statistical analysis was performed using a one-way ANOVA, followed by a Tukey test. Significant differences (*p* < 0.05) between particle-conjugated antigens and unconjugated particles are represented with #.

**Table 1 pharmaceuticals-15-00887-t001:** Conjugation efficiencies of the conjugates used throughout the study.

	Unlabeled 500 nm Particles	Unlabeled 3 µm Particles	Fluorescently Labeled 500 nm Particles	Fluorescently Labeled 3 µm Particles
Unlabeled Influenza antigen	1.25 × 10^−9^ µg/µm^2^	6.90 × 10^−9^ µg/µm^2^	1.29 × 10^−9^ µg/µm^2^	4.16 × 10^−9^ µg/µm^2^
Unlabeled HBsAg	0.926 × 10^−9^ µg/µm^2^	4.69 × 10^−9^ µg/µm^2^	-	-
Fluorescently Labeled influenza antigen	0.410 × 10^−9^ µg/µm^2^	0.196 × 10^−9^ µg/µm^2^	-	-
Fluorescently Labeled HBsAg	1.31 × 10^−9^ µg/µm^2^	6.91 × 10^−9^ µg/µm^2^	-	-

**Table 2 pharmaceuticals-15-00887-t002:** Slope differences between uptake of particle-conjugated antigens versus unconjugated antigens.

Conjugate Type	Slope of Uptake	Fold-Change in Slope Compared to Free Antigen
3 µm-influenza antigen	94,027	18.4
500 nm-influenza antigen	45,899	1.2
3 µm-HBsAg	310,806	2.6
500 nm-HBsAg	622,956	4.2

**Table 3 pharmaceuticals-15-00887-t003:** Stimulation scheme for the quantitative uptake studies in monocyte-derived dendritic cells.

Conjugate Type	Number of Particlesper Well	Amount of Corresponding Antigen per Well
3 µm-influenza antigen	20 µg	6.4 ng
500 nm-influenza antigen	20 µg	93.8 ng
3 µm-HBsAg	20 µg	248.6 ng
500 nm-HBsAg	20 µg	298.7 ng

## Data Availability

The data presented in this study are available in article or Appendix A.

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
