# Peer review of "Assessing the Immunomodulatory Effect of Size on the Uptake and Immunogenicity of Influenza- and Hepatitis B Subunit Vaccines In Vitro"

_pharmaceuticals, 2022, doi:10.3390/ph15070887_

Round 1
Reviewer 1 Report
In this manuscript, the author prepared antigen-conjugated polystyrene particles having different sizes, 500 nm and 3 µm, and compared their cellular uptake and capacity to induce T cell activation using human MoDCs and PBMCs. This study demonstrated the conjugation of antigens to polystyrene particles increased their uptake by MoDCs. In addition, the study also demonstrated influenza-antigen conjugated particles stimulated cytokine production by CD4 T cells on the other hand HBsAg conjugated particle did not stimulate cytokine production by T cells. However, the author’s discussions in the paper did not align with the initial objectives. Without more rigorous testing of particle uptake and immune response-induction or more precise discussion about the result, the paper is not a good fit for pharmaceuticals.
Major points
1. The author showed uptake of influenza antigen-conjugated particles by MoDCs in Figure 4 but not those of HBsAg conjugated particles. Did author check the HBsAg conjugated particles also taken up by MoDCs?
2. In this study, the author prepared fluorescently labeled antigen conjugated particles. However, the conjugation efficiency of fluorescently labeled influenza antigens were very low and therefore, there were some differences in experimental condition between Figure 5 and Figure 6,7. Is this because FITC-labeling reduced free amine groups on antigens which are needed for the conjugation to particle? If so, it is better to label the antigens in other way, for example labeling sulfhydryl group on antigens, and make the experimental condition all the same throughout the study.
3. The initial purpose of this study is to determine whether particle size affect the cellular uptake and following induction of immune responses. However, the author emphasized that capacity of conjugates to induce T cell stimulation was dependent on the immunological status of the donors. It seems that the author’s discussion did not align with the initial objectives. If it is possible, it is better to use another immunogenic model antigens that is not experienced by human, for example keyhole limpet hemocyanin, to confirm the effect of the particle sizes without affected by donor’s immunological status. Or it is better to add another possible explanations for Figure 6 and 7 that is more related to particle sizes and particle uptake.
Minor points
1. There is an extra space in line 36 between “more” and “antigens”.
2. In line 357, the author should declare the location (city) of Polysciences. In line 377, the author do not have to declare the company information of Polysciences as it is the second time appeared in the paper.
Author Response
Response to reviewer 1
Comments and Suggestions for Authors
In this manuscript, the author prepared antigen-conjugated polystyrene particles having different sizes, 500 nm and 3 µm, and compared their cellular uptake and capacity to induce T cell activation using human MoDCs and PBMCs. This study demonstrated the conjugation of antigens to polystyrene particles increased their uptake by MoDCs. In addition, the study also demonstrated influenza-antigen conjugated particles stimulated cytokine production by CD4 T cells on the other hand HBsAg conjugated particle did not stimulate cytokine production by T cells. However, the author’s discussions in the paper did not align with the initial objectives. Without more rigorous testing of particle uptake and immune response-induction or more precise discussion about the result, the paper is not a good fit for pharmaceuticals.
First of all, we would kindly like to thank reviewer 1 for the constructive criticism provided in relation to our manuscript, especially with regard to the discussion section. We have now revised the discussion section in light of our initial objectives as suggested by the reviewer. The changes we made are specified under comment number 3, below.
Major points
- The author showed uptake of influenza antigen-conjugated particles by MoDCs in Figure 4 but not those of HBsAg conjugated particles. Did author check the HBsAg conjugated particles also taken up by MoDCs?
What we envisioned to show here, basically, was if there is any visual difference in the uptake of polystyrene particles, either conjugated to antigens or unconjugated, by monocyte-derived dendritic cells (Mo-DCs). We did this in order to get a first impression of particle uptake in general. For the sake of hepatitis B surface antigen- (HBsAg) scarcity, we decided to use here only the influenza conjugates in order to answer this question. Moreover, as the results demonstrated that plain and antigen-conjugated particles were taken up to a similar extent, we did not expect to see any (visual) differences in uptake pattern when HBsAg-conjugated particles were to be used. Therefore, we decided to stick with the influenza conjugates for visualization purposes, while instead using both influenza antigen- and HBsAg-conjugated beads for quantification by flow cytometry.
- In this study, the author prepared fluorescently labeled antigen conjugated particles. However, the conjugation efficiency of fluorescently labeled influenza antigens were very low and therefore, there were some differences in experimental condition between Figure 5 and Figure 6,7. Is this because FITC-labeling reduced free amine groups on antigens which are needed for the conjugation to particle? If so, it is better to label the antigens in other way, for example labeling sulfhydryl group on antigens, and make the experimental condition all the same throughout the study.
First of all, we agree with the reviewer that the conjugation efficiency of FITC-labeled influenza subunit (as depicted in Table 1) was low and therefore did not match the conjugation efficiency found for the unlabeled conjugates used in the immune response experiments (Figure 6). As the FITC-labeling method is directed at primary amines from lysine residues present in the amino acid-sequence of the protein, it may indeed have been the case that the FITC labeling compromised the free amines needed for particle conjugation, suggesting that the labeling might have affected protein conjugation efficiency. However, we did not see the compromised conjugation efficiencies for HBsAg, indicating that a putative effect of the FITC labeling would be protein specific.
Given this observed variation in conjugation efficiency for particles of different sizes and for the different proteins, we decided to compare the uptake of particles on basis of their uptake rate only rather than on basis of the achieved level of fluorescence. In this way, we could compare the uptake rate for the same amount of antigen either free or conjugated. Therefore, we argue that the differences in conjugation efficiency do not affect the conclusions.
- The initial purpose of this study is to determine whether particle size affect the cellular uptake and following induction of immune responses. However, the author emphasized that capacity of conjugates to induce T cell stimulation was dependent on the immunological status of the donors. It seems that the author’s discussion did not align with the initial objectives. If it is possible, it is better to use another immunogenic model antigens that is not experienced by human, for example keyhole limpet hemocyanin, to confirm the effect of the particle sizes without affected by donor’s immunological status. Or it is better to add another possible explanations for Figure 6 and 7 that is more related to particle sizes and particle uptake.
We acknowledge that our explanation about the association between antigen size and T cell stimulation lacked clarity and tried to rectify this by amending the text accordingly (lines 274-307). The most likely explanation for the fact that the enhanced antigen uptake through conjugation to particles observed for both antigens translated to improved T cell responses for the influenza antigen but not for HBsAg is the following: Due to frequent exposure to influenza virus, all donors possess memory T cells which require for stimulation relatively low levels of T cell receptor engagement and no co-stimulatory signals (reference in revised manuscript). Accordingly, enhanced presentation of particle-conjugated antigen alone resulted in enhanced T cell stimulation. In contrast, our donors were most likely naïve for HBsAg. Activation of naïve T cells requires more effective antigen presentation and in particular effective co-stimulatory signals. While coupling of the antigen to beads did improve antigen uptake and thus presumably antigen presentation, it was unable to provide the required co-stimulatory signals.
Although using different antigens (like the suggested keyhole limpet hemocyanin) could indeed control for the influence of immunological memory, using such antigens would not be of use within the broader context of particulate delivery systems for human directed subunit vaccines. Moreover, we feel we have controlled for such effects already by including both plain antigens and unconjugated particles in the experiments. For this reason, a difference in immune response upon stimulation with conjugates versus plain antigen would be a direct effect of the conjugation and thus, of the increased size. Although we did not see big differences in immune responses upon stimulation with conjugates versus plain antigens, the difference was visible for the uptake rates (where conjugation led to more efficient uptake). Concluding, size clearly had an effect on uptake of both influenza and HBsAg conjugates, while the effect on the immune response likely depended on the immunological status of the donors.
Minor points
- There is an extra space in line 36 between “more” and “antigens”.
We have removed the extra space and thank the reviewer for noticing.
- In line 357, the author should declare the location (city) of Polysciences. In line 377, the author do not have to declare the company information of Polysciences as it is the second time appeared in the paper.
This has been changed now.
Reviewer 2 Report
Assessing the immunomodulatory effect of size on the uptake and immunogenicity of influenza and hepatitis B subunit vaccines in vitro.
The author explained in detail the effect of the particle size on the ability of the vaccine. I was able to understand the explanation easily, but I would like to ask you a few questions about the manuscript.
1. The polystyrene particles synthesized by the author do not appear to be biodegradable at all in Figure 4. Studies have also shown that polystyrene nanoparticles are toxic to mice (ref: Xu, D., Ma, Y., Han, X., & Chen, Y. (2021). Systematic toxicity evaluation of polystyrene nanoplastics on mice and molecular mechanism investigation about their internalization into Caco-2 cells. Journal of Hazardous Materials, 417, 126092.). Therefore, the author expects better research results if we conduct a biodegradability and toxicity assessment of Mo-DCs of polystyrene particles.
2. The nm size has higher absorption than the um size. Comparing with the absorption rate of pure particles that do not conjugate submit and Antigen in part 2.4, it seems reliable.
3. In reference to Figure 1, fluorescently labeled particle and fluorescently labeled antigen and unlabeled particle were put as controls, and I think it would have been better if both antigens and particles could have added fluorescently labeled particles as controls and checked the results at the same time.
4. In Introduction 54-59, it is said that human-derived dendritic cells become more representative immune system models than animal models. Here, I am curious about the difference in mechanism between animal models and human-derived dendritic cells.
5. If you can express in more detail what the p-value compares in the graphs of Figure 6 and Figure 7, it will be easier for readers to understand.
6. In Figure 4, it would be good to indicate what dye was used for the nucleus. And It would also be nice if there was a comment on why the cells were backwashed for 10 days in Figure 7
7. In Figure 3, zeta potential, it seems necessary to explain why the zeta potential increased after conjugation with influenza subunits or HBsAg compared to amine-functionalized polystyrene particles and their meaning.
8. Comparing (c) and (d) in Figure 4, it was found that more nanoparticles entered the cells in the influenza subunit vaccine conjugated particles (500 nm) on the image, indicating higher uptake efficiency than the unconjugated fluorescently labeled particles (500 nm). In main text 2.3, is there any reason why you said that there was no quantitative difference in uptake between the four groups?
Author Response
Response to reviewer 2
Comments and Suggestions for Authors
Assessing the immunomodulatory effect of size on the uptake and immunogenicity of influenza and hepatitis B subunit vaccines in vitro. The author explained in detail the effect of the particle size on the ability of the vaccine. I was able to understand the explanation easily, but I would like to ask you a few questions about the manuscript.
We would like to thank the reviewer for the comments given and have elaborated on this below.
- The polystyrene particles synthesized by the author do not appear to be biodegradable at all in Figure 4. Studies have also shown that polystyrene nanoparticles are toxic to mice (ref: Xu, D., Ma, Y., Han, X., & Chen, Y. (2021). Systematic toxicity evaluation of polystyrene nanoplastics on mice and molecular mechanism investigation about their internalization into Caco-2 cells. Journal of Hazardous Materials, 417, 126092.). Therefore, the author expects better research results if we conduct a biodegradability and toxicity assessment of Mo-DCs of polystyrene particles.
The reviewer is absolutely right, but we certainly do not envision to administer polystyrene particles as future carrier for delivering vaccines in vivo. We are fully aware that polystyrene is not biodegradable and therefore not suitable for in vivo studies as it indeed may lead to toxicity. We used here polystyrene particles simply because they are well defined and highly monodisperse. Also, they are available with many functional groups that enables the covalent linkage of antigens. Therefore, these (fluorescent) particles are a robust and easy to use model for studying the fundamental questions on the influence of size on particle uptake and T cell stimulation. Notwithstanding, aware of the pitfalls of using polystyrene, already in lines 338-342 and 349-352 we hint at translating the work from this study to biodegradable and biocompatible particles to see potential differences and study the long-term effects. An additional sentence (Line 338-339) was added underlining potential toxicity.
- The nm size has higher absorption than the um size. Comparing with the absorption rate of pure particles that do not conjugate submit and Antigen in part 2.4, it seems reliable.
We thank the reviewer for this mention.
- In reference to Figure 1, fluorescently labeled particle and fluorescently labeled antigen and unlabeled particle were put as controls, and I think it would have been better if both antigens and particles could have added fluorescently labeled particles as controls and checked the results at the same time.
We are not sure whether we fully understand the point raised here. Notwithstanding, we provide answers to two possible interpretations of the comment. A first interpretation could be whether it would have been better to include fluorescently labeled particles as controls throughout the whole study.
For the immune response experiments, fluorescently labeling the particles (or antigens) was not necessary as our primary read-out was intracellular cytokine expression. For the microscopy experiments we actually needed fluorescently labeled particles as a control in order to be able to visualize plain unconjugated particles. For studying the uptake rates by flow cytometry, we decided to use fluorescently labeled antigens as a control as we were primarily interested in the influence of particle conjugation on uptake of the antigens (either free or conjugated to particles), instead of the influence of antigens on the uptake of particles.
Another interpretation of the reviewer’s comment could be whether it would have been better to have two colors, 1 for the particles and 1 for the antigen, to be able to measure both particle and antigen uptake simultaneously. We decided to avoid double labelling to exclude interference of the 2 fluorophores and their emission which could complicate interpretation of the results. Instead we used different labelled conjugates depending on the scientific question. Firstly, we used conjugates with labelled particles to visualize particle uptake with and without antigen (Figure 4) and secondly, we used conjugates with labelled antigens to quantify antigen uptake efficiency (Figure 5).
- In Introduction 54-59, it is said that human-derived dendritic cells become more representative immune system models than animal models. Here, I am curious about the difference in mechanism between animal models and human-derived dendritic cells.
We would like to point out that, of course, a cell model is not at all representative for a whole animal, as it cannot model the interplay of cells (and their environment) within a complex tissue (let alone a living organism) and neglects the metabolic processes that occur in vivo. To that extent, it is not possible to compare a whole animal model with a single cell model (or even a cell line). We believe that this point in general holds true for any type of in vitro research on cell cultures, but the controllable nature of such in vitro cultures makes it indispensable for fundamental research.
Specifically, for studying the questions with regard to particle uptake in cells, the human-derived Mo-DC model is a well-suited alternative to animal models. Moreover, in vitro cell cultures contribute to reduction of animals needed for such research and dendritic cells are a key player in initiation of antigen-specific immune responses. In light of the immune response experiments, we have deliberately chosen for the peripheral blood mononuclear cell (PBMC) model, as this harbors the vast majority of immune cells present in the blood (including dendritic cells) and allows for cellular interactions within the culture (e.g. between antigen-presenting cells and T cells).
- If you can express in more detail what the p-value compares in the graphs of Figure 6 and Figure 7, it will be easier for readers to understand.
We have changed the text in the caption of Figures 6 and 7 on the definition of the p-value to read ‘particle-conjugated antigens’ instead of ‘-vaccines’ and ‘unconjugated particles’ instead of ‘plain particles’.
- In Figure 4, it would be good to indicate what dye was used for the nucleus. And It would also be nice if there was a comment on why the cells were backwashed for 10 days in Figure 7
We have now added the dyes to the caption of Figure 4. In Fig 6 and 7, the PBMCs were cultured for 10 days with the antigens in order to allow antigen-specific T cells to proliferate and differentiate into cytokine-producing cells. We have added this information to the text (218-219).
- In Figure 3, zeta potential, it seems necessary to explain why the zeta potential increased after conjugation with influenza subunits or HBsAg compared to amine-functionalized polystyrene particles and their meaning.
it is important to note that despite the presence of the amine group, the amino-functionalized particles displayed a negative zeta potential in the conditions tested (in PBS, pH 7.4). After conjugation using the glutaraldehyde reaction, we saw a decrease in zeta potential (absolute value) towards neutrality. This suggests that the conjugation reduces the charge (in absolute value). Overall, the change of zeta potential is an indication that the conjugate was successfully formed. To clarify, we have changed the notion of ‘zeta potential increase’ into the notion that the particles became ‘less negatively charged’ (Lines 121-122) and have included in the caption that the particles were stored in PBS, pH 7.4 (Line 134).
- Comparing (c) and (d) in Figure 4, it was found that more nanoparticles entered the cells in the influenza subunit vaccine conjugated particles (500 nm) on the image, indicating higher uptake efficiency than the unconjugated fluorescently labeled particles (500 nm). In main text 2.3, is there any reason why you said that there was no quantitative difference in uptake between the four groups?
We would like to stress here that in figure 4, describing the microscopy experiments, we are zooming in on just a few cells to gain a qualitative impression on how cells handled the particles. For this reason, it may be that a particular cell that is imaged contains more particles than a neighboring cell, making it not representative for the whole cell population present in the culture. In this light, these pictures can in no way be used as a quantitative measure for assessing particle uptake per cell. We have done this instead in figure 5. However, we feel we should have stated this differently in the text, as we indeed used the term ‘quantitative’. We have clarified this by changing lines 136-137 and by removing the sentence in lines 143-144.
Round 2
Reviewer 1 Report
I understand the authors explanations about the experimental conditions and the discussion.
However, the authors should address the following point about the discussion before publication. There is also an minor points the author should correct.
- I understand the author’s discussion written in lines 274-307. However, the sentence in lines 303-307 might be overstatement since the co-stimulatory signals were not checked in the paper. The author should alter the sentence to express the possibility appropriately.
- After the revision, the author changed the word “(subunit) vaccines” into “antigens”, such as in lines 148, 237, and etc. However there are still some points the author describes the “antigens” as “vaccines”, such as in lines 103-105, 111-112, and etc. The author should use the same terminology throughout the manuscript and also in the graphs.
Author Response
Response to reviewer 1 round 2
I understand the authors explanations about the experimental conditions and the discussion.
However, the authors should address the following point about the discussion before publication. There is also an minor points the author should correct.
We thank the reviewer kindly for agreeing on our explanations and for providing some additional suggestions for revision.
- I understand the author’s discussion written in lines 274-307. However, the sentence in lines 303-307 might be overstatement since the co-stimulatory signals were not checked in the paper. The author should alter the sentence to express the possibility appropriately.
We agree with the reviewer on this point and therefore have added the words ‘we believe’ to the respective sentence (316) to underline that it is based on a hypothesis. Also, we have added an additional sentence from line 318 as a suggestion for future studies.
- After the revision, the author changed the word “(subunit) vaccines” into “antigens”, such as in lines 148, 237, and etc. However there are still some points the author describes the “antigens” as “vaccines”, such as in lines 103-105, 111-112, and etc. The author should use the same terminology throughout the manuscript and also in the graphs.
We agree with the reviewer that we have not been consistent in terminology. Therefore, we have now changed these terms throughout the manuscript to make them consistent. Also, in Figure 5 the term ‘unconjugated vaccine’ has been changed into ‘unconjugated antigens’. In lines 68-69 we have included in-between brackets that the influenza antigens are derived from subunits as we feel it is important to underline that they were purified from whole inactivated virus (as stated in the methods section).